# Automatic Measurement and Characterization of the Dynamic Properties of Tethered Membrane Wings

Jan Hummel[1], Dietmar Göhlich[1], and Roland Schmehl[2]

[1]Methods for Product Development and Mechatronics, Technische Universität Berlin, 10623 Berlin, Germany
[2]Faculty of Aerospace Engineering, Delft University of Technology, 2629 HS Delft, Netherlands

**Correspondence:** Jan Hummel (jan.hummel@tu-berlin.de)

**Abstract.** We have developed a tow test setup for reproducible measurement of the dynamic properties of different types of tethered membrane wings. The test procedure is based on repeatable automated maneuvers with the entire kite system under realistic conditions. By measuring line forces and line angles, we determine the aerodynamic coefficients and the lift-to-drag ratio as functions of the length ratio between power and steering lines. This non-dimensional parameter characterizes the angle-of-attack of the wing and is varied automatically by the control unit on the towed test bench. During each towing run, several test cycles are executed such that mean values can be determined and errors can be minimized. We can conclude from this study that an objective measurement of specific dynamic properties of highly flexible membrane wings is feasible. The presented tow test method is suitable for quantitatively assessing and comparing different wing designs. The method represents an essential milestone for the development and characterization of tethered membrane wings as well as for the validation and improvement of simulation models. On the basis of this work, more complex maneuvers and a full degree of automation can be implemented in subsequent work. It can also be used for aerodynamic parameter identification.

## 1 Introduction

With the turn of the millennium, kitesurfing has evolved into a mainstream water sport, followed by the more recent variants of land and snow kiting (Tauber and Moroder, 2013). In terms of industrial applications, flexible membrane wings have already been used since the 1970s as aerodynamic decelerators for airdrop systems and are currently being explored for airborne wind energy (AWE) generation (Schmehl, 2018). Despite the advancements within the kitesurfing and AWE industries, tethered membrane wings are mostly still designed by iterative testing with empirical and intuitive variation of wing parameters. Although this has led to a relatively high degree of maturity on product level, the approach is time consuming and expensive because a large number of prototypes need to be manufactured and tested. For this reason, we conclude that the empirical design method will allow only limited further improvements and that it is indispensable to develop a systematic understanding of how wing performance parameters such as aerodynamic coefficients, lift-to-drag ratio, steering forces and moments depend on the wing design.

The empirical design method is used because compared to rigid wings the physics of flexible membrane wings is complex and the existing knowledge is limited, due to deforming under aerodynamic load and steering line actuation. This holds particu-

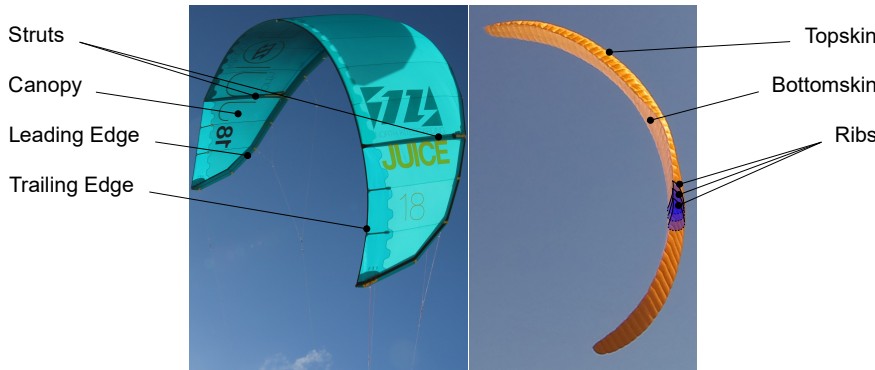

**Figure 1.** left: Leading Edge Inflatable (LEI) tube kite; right: ram air wing

larly for Leading Edge Inflatable (LEI) tube kites (see Fig. 1) and other single-skin kite types, since ram air wings have already been investigated systematically for several decades (Dunker, 2013; Johari et al., 2014; Dunker, 2018). Because of the high degree of flexibility and the low weight of the membrane structure, the flow around the wing and its shape are strongly coupled. A change in the flow field alters the aerodynamic load distribution to which the structure rapidly adjusts by deformation, which
in turn changes the flow field. The fluid-structure coupling cause deformation phenomena at different length and time scales (Leuthold, 2015). A typical large-scale phenomenon is the spanwise bending and twisting of the entire wing due to steering line actuation. The ability of the membrane wing to deform asymmetrically and by that generate a substantially increased turning moment makes it particularly suitable for AWE applications which require excellent maneuverability (Breukels et al., 2013; Bosch et al., 2014; van Reijen, 2018; Fechner and Schmehl, 2018). Typical small-scale phenomena are the local flutter of the
wing canopy or wrinkling, which is caused by local compression loads that can not be supported by the woven fabric material.

Another characteristic that distinguishes flexible membrane wings from rigid wings is that the entire airborne system, consisting of wing, tensile support system and in some cases also a suspended airborne control unit, is considerably larger for comparable traction force. This is due to the fact that a rigid wing can endure a much higher wing loading than a membrane wing and that it uses aerodynamic control surfaces with wing-integrated actuators which allow a more compact design. For
wind tunnel measurements large geometries are typically downscaled to fit into the test section of the tunnel. To ensure that the flow field is not affected by the scaling, the principle of dynamic similarity has to be enforced by maintaining a constant Reynolds number $\mathrm{Re} = \rho v c / \mu$. A common method to compensate a decreasing chord length $c$ is to increase the flow velocity $v$. However, downscaling a tethered membrane wing for wind tunnel testing is problematic, because due to aeroelasticity the aerodynamic characteristics depend not only on the wing geometry but also on its deformation behavior. To account for this,
the material properties of the wing and tether would have to be scaled accordingly, which is practically not feasible because the membrane is a woven fabric material, that is partially arranged as multilayer composite and with rigid reinforcements, and the tether is a braided and coated line (Bosman et al., 2013).

A wind tunnel study of a small but full-scale ram air wing was presented by de Wachter (2008). The wing with a projected area of 5.2 m$^2$ was suspended upside down in the test sections of two different large wind tunnels to determine the shape under

aerodynamic load by photogrammetry and laser scanning. This shape was then used as static boundary condition for steady CFD analysis, with the aim to assess the computational prediction quality without the added complexity of the deforming membrane structure. The study contributed important knowledge about ram air wings at the lower end of the size range. In the same framework project, Bungart (2009) performed a coupled CFD and finite element analysis of a ram air wing section,

deriving aerodynamic coefficients and deformed shape for the entire range of angle of attack. The analysis showed that the chambered design (chambers are separated by ribs, topskin and bottomskin, see Fig. 1) with upper and lower skin and the airfoil defined by a small number of ribs (connecting topskin and bottomskin) leads to ballooning. A similar effect can be observed with LEI tube kites, where the canopy is bulging out between the struts (inflatable tube providing structure) that similar to the ribs define the design shape. It is obvious, that these aeroelastic phenomena have to be taken into account by

high-fidelity analyses. Subsequently, Breukels (2011) developed a multibody model and Bosch et al. (2014) a finite element model of the flexible wing, bridle line system (line system which supports the wing structure and merges these lines into steering or power lines) and tether. In both approaches the same correlation derived by parametric CFD analysis is used to evaluate the aerodynamic load distribution as a function of angle of attack and wing deformation. While succeeding in simulating complete flight maneuvers relevant for AWE, the two studies did not include validations by wind tunnel experiments. It can be concluded

that validated aeroelastic models of entire tethered membrane wings are neither available at present, nor will they be sufficiently fast to be used in the design process where rapid iterations are required.

For this reason, less complex simulation models have been developed, describing the whole kite system as a point mass, a cluster of point masses (Fechner et al., 2015) or a rigid body (de Groot et al., 2011; Gohl and Luchsinger, 2013). These models do not explicitly describe the aeroelastic behavior of the wing and require as input the detailed aerodynamic properties

of the kite system, including information about the steering behavior. In this respect, Erhard and Strauch (2013a, b); Fagiano et al. (2014) and Jehle and Schmehl (2014) have proposed empirical turn rate laws relating the turn rate of the wing to the steering input. The transition from powered state (high angle of attack) to depowered state (low angle of attack) is covered by an empirical correlation (Fechner et al., 2015). According to Fagiano and Marks (2015), such lower complexity models have already reached a quite mature state, but new insights appear to be difficult to gain without experimental analysis.

However, despite the strong need for reproducible experimental data, only few dedicated studies have been performed so far. Stevenson (2003) developed a tow test method to support the research and development of surf kites. The constant relative airflow was generated by driving the towing vehicle along a beach section. The data acquisition system recorded the lift-to-drag ratio as well as the lift coefficient both as functions of the ratio of the sum of steering line forces to total tensile force. Inspired by a method described by Stevenson et al. (2005), a simple stationary test setup for the beach was used by van der Vlugt (2010)

to determine the lift-to-drag ratio of surf kites from the achievable flight speed when performing crosswind sweeps close to the ground. Dadd et al. (2010) described a tow test with the measurement rig mounted on a trailer such that it could be used for stationary as well as for tow testing. A tow test experiment for characterization of kites used as part of an AWE system was described by Costa (2011). Next to the movement of the kite and the line forces also the deformation was measured, using an image correlation system. Within the same framework project, Wood et al. (2017) presented a control strategy for flying

figure-of-eight crosswind maneuvers during tow tests.

In none of the outlined test procedures the manual control input was recorded. However, for systematic aerodynamic parameter identification a recording of the steering inputs is crucial (de Groot et al., 2011; Mulder et al., 1994). We started the project TETA at TU Berlin with the aim to measure the dynamic properties of kites under reproducible conditions for repeatable steering input (Hummel, 2017; Hummel and Göhlich, 2017). The developed test setup is suitable for quantitative assessment

of different types of tethered membrane wings and can be used stationary or moving at variable velocity to simulate different wind speeds as well as to reduce the influence of gusts.

This paper is organized as follows. Section 2 outlines the measurement concept and describes the details properties to be measured. In Sect. 3 the setup and design of the test bench is described, including the required sensor equipment. Section 4 continues with a brief overview of the data acquisition process. In Sect. 5 the experimental results are presented and discussed.

In the conclusions, future research and improvements of the measurement concept and the implemented test bench are outlined.

## 2   Measurement Concept

A schematic side view of the tow test is illustrated in Fig. 2, including the aerodynamic lift and drag force components $\mathbf{F}_L$ and $\mathbf{F}_D$, respectively, as well as the aerodynamic force $\mathbf{F}_A$. The resultant force acts in the center of pressure of the wing. A steady

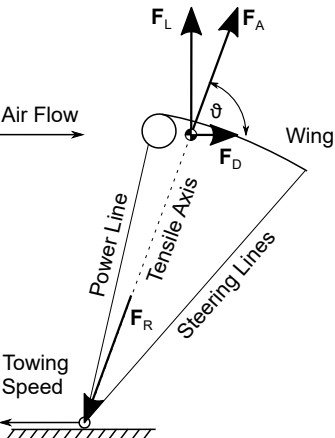

**Figure 2.** Schematic side view of the tow test with the wing in steady state equilibrium and effect of gravity neglected ($\mathbf{F}_R + \mathbf{F}_A = 0$ with $m\mathbf{g} = \mathbf{0}$ and $\mathbf{F}_R = \mathbf{F}_{PL} + \mathbf{F}_{SL,l} + \mathbf{F}_{SL,r}$)

towing state is reached when the wing is not moving anymore relative to the towing vehicle. In this state, the aerodynamic

and gravitational forces acting on the wing are balanced by the tensile forces $\mathbf{F}_{PL}$, $\mathbf{F}_{SL,l}$ and $\mathbf{F}_{SL,r}$ acting in the power and steering lines. Because flexible lines can not support bending loads these tensile forces are always aligned with the lines, as illustrated in Fig. 3. The dashed line in Fig. 2 defines the tensile axis of the airborne system, which in case of a negligible effect of gravity is aligned with the resultant force $\mathbf{F}_R$ and inclined to the horizontal plane by the elevation angle $\vartheta$.

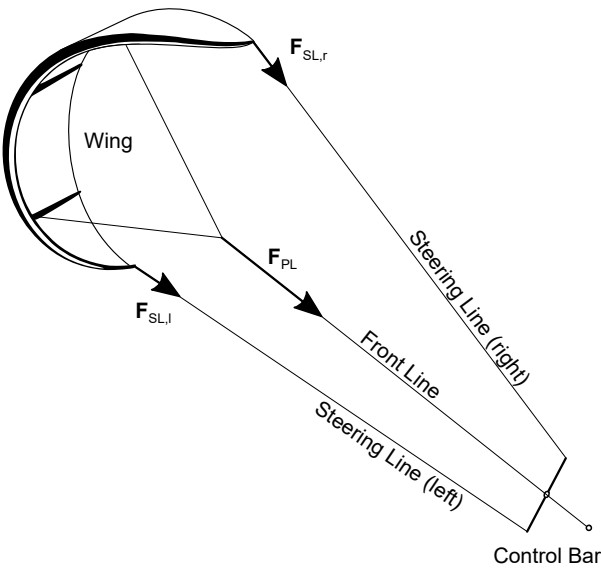

**Figure 3.** Forces acting in the power and steering lines

As illustrated in Fig. 3, the power line is attached to the towing point at the moving test rig. The flight behavior of the wing is controlled by a bar which can slide along the power line and attaches at its ends to the two steering lines. This setup is commonly used for kitesurfing and allows individual actuation of the left and right steering lines and changing the effective length of the power line. The effective length of the power line is defined as the distance between the kite attachment point and

the control bar. For the "Linear Power" maneuver the control bar is automatically retracted along the power line. During this maneuver the effective length $l_{PL}$ of the power line changes from $l_{PL,0}$ for the depowered state to $l_{PL,1}$ for the powered state, as illustrated in Fig. 4. Accordingly we define the relative power setting

$$u_p = \frac{l_{PL} - l_{PL,0}}{l_{PL,1} - l_{PL,0}}, \tag{1}$$

which varies between $u_p = 0$ for the depowered state and $u_p = 1$ for the powered state. A similar nondimensional variable, the

relative depower setting $u_d = 1 - u_p$, was introduced by Fechner et al. (2015) to quantify the actuation of an airborne control unit suspended below the wing.

In the following we describe the wing properties that are used to characterize the flight dynamic behavior of the wing. In order to facilitate an easy assessment of the measurement results as well as the reliability of the method, post-processing calculations to optimize the estimation of the properties were not carried out. Since kites were tested at same wind speed, of

same size and control bar settings, a relative comparison of the wings is still possible.

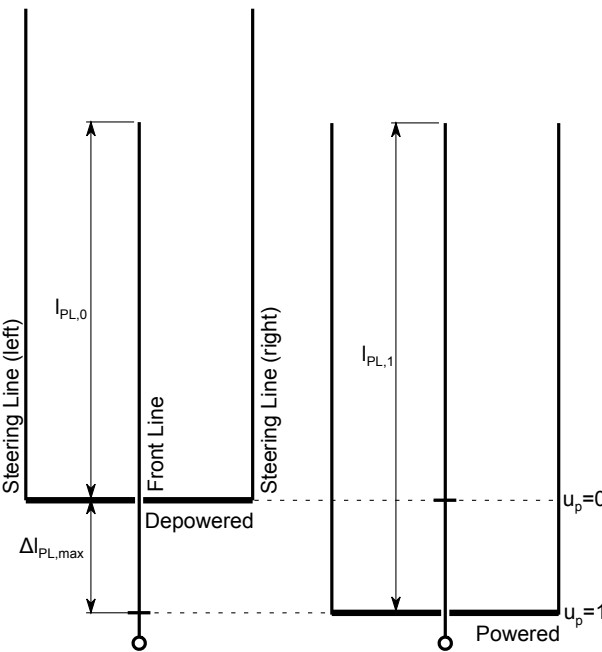

**Figure 4.** Limiting states of the "Linear Power" maneuver

## 2.1 Aerodynamic Coefficients

The aerodynamic coefficients are non-dimensional parameters that describe the aerodynamic properties of a wing. For a steady towing situation as illustrated in Fig. 2 we can determine the lift, drag and resultant aerodynamic coefficients of the entire system as

$$C_L = \frac{2F_L}{\rho A v^2} = \frac{2\sin\vartheta F_R}{\rho A v_a^2}, \tag{2}$$

$$C_D = \frac{2F_D}{\rho A v^2} = \frac{2\cos\vartheta F_R}{\rho A v_a^2}, \tag{3}$$

$$C_R = \frac{2F_R}{\rho A v_a^2}, \tag{4}$$

where $\rho$ is the air density, $A$ the surface area of the wing and $v_a$ the apparent wind velocity. By definition the aerodynamic drag is aligned with the apparent wind velocity, the aerodynamic lift is perpendicular.

Based on the resultant aerodynamic force coefficient we can determine the depower capability of the wing. This parameter can be calculated as relative difference of maximum and minimum aerodynamic forces

$$\gamma = \frac{C_{R,max} - C_{R,min}}{C_{R,max}}, \tag{5}$$

evaluating the entire range $1 > u_p > 0$. For ground-generation AWE systems it is the traction force of the kite that is converted into electricity (Schmehl et al., 2013). For this variant of the technology, the kite is generally operated in consecutive pumping

cycles and for maximizing the energy output, the resultant force coefficient $C_R$ has to be maximized during the traction phases and minimized during the retraction phases. For a flexible membrane wing, a good depower capability and flight stability are two conflicting design drivers (van der Vlugt et al., 2013).

## 2.2 Aerodynamic Efficiency

The aerodynamic efficiency of a wing can be expressed as the ratio between the aerodynamic lift and drag force components. For a steady towing situation as illustrated in Fig. 2 the lift-to-drag ratio can be calculated from the elevation angle $\vartheta$ as

$$\frac{F_L}{F_D} = \frac{C_L}{C_D} = \tan \vartheta. \tag{6}$$

The lift-to-drag ratio is also a measure for the achievable flight speed of the kite in crosswind motion (van der Vlugt, 2010; Schmehl et al., 2013).

## 2.3 Tether Forces

The tensile forces acting in the power and steering lines are shown in Fig. 3. The ratio of the steering line forces to the power line force

$$f = \frac{F_{SL,l} + F_{SL,r}}{F_{PL}} \tag{7}$$

characterizes the load distribution between the rear and front parts of the tethered wing, which allows the validation of simulation approaches. Additionally, in order to characterize sport kites this parameter was used so far intuitively to describe the perceived steering forces. Hence, a quantitative comparison of different wings regarding the load distribution between power and steering lines is feasible.

## 3 Test Bench Setup

The following section gives a brief overview of the developed test bench. The main design goals are as follows: (1) using the entire kite system (including the unscaled kite and tether as well as the common steering input device) to generate realistic measurement data, (2) providing constant and controllable flow conditions, (3) allowing repeatable and automated steering inputs, (4) as little as possible of an impairment to the wing and its control unit by attachments, (5) ensuring an easy transport and tow of the test bench. The final version of the test bench is shown in Fig. 5 and the schematic principle is illustrated in Fig. 6.

## 3.1 Structural Design

With regard to the acquisition costs of the towed platform, a permanent mounting on a car trailer was decided. This solution allows to use any given car for towing and thereby avoid additional costs. However, in contrast to heavier vehicles (e.g. four wheel vehicles with a driver's cab), the influence of oscillations into the test bench by the tethers is expected. This results in

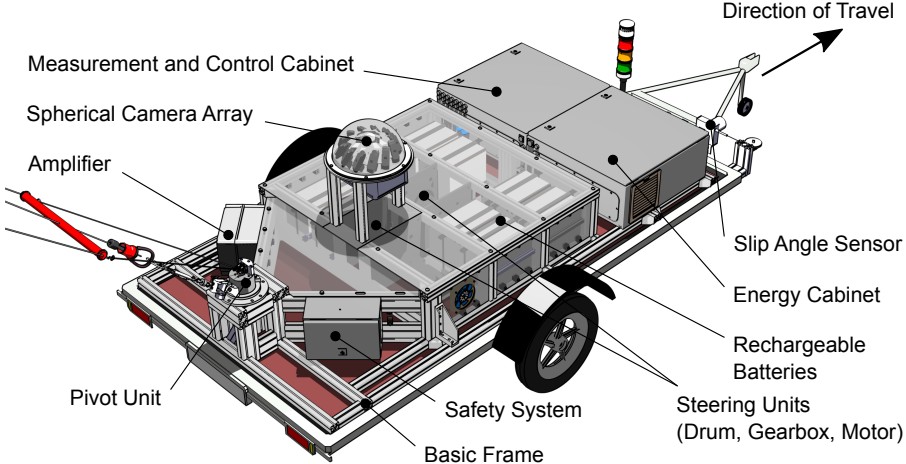

**Figure 5.** Components of the trailer-mounted test bench

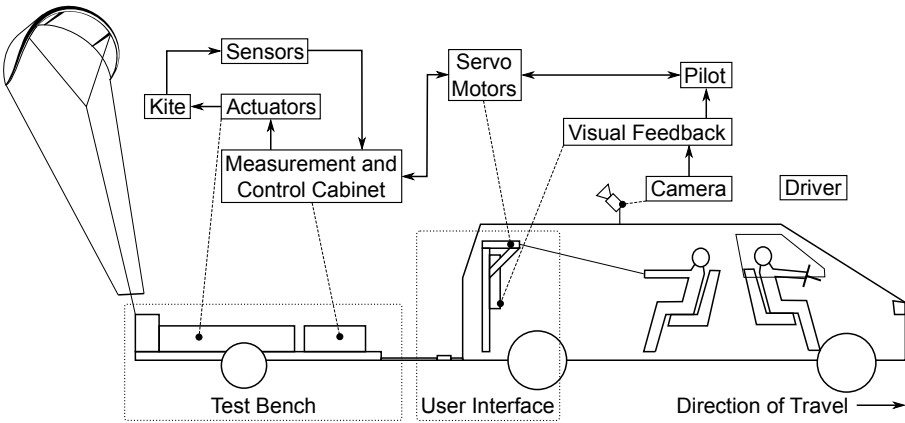

**Figure 6.** Towing test schematic

an additional requirement for the design of the test bench. All components are connected in such a way that it is possible to change the driving platform in the future to further improvements. For example, vibrations induced by the single-axle trailer could be greatly reduced by mounting the test rig on a heavier platform.

The basic frame is used to mount the test bench modules and absorb the load, in particular the line forces. It is assembled from aluminum profiles, to avoid corrosion and easily afford subsequent design modifications. The kite is connected to the test bench by the pivot unit, which is located in the rear of the trailer (in relation to the direction of travel).

The pivot unit is shown in Fig. 7. It is designed to have a minimum inertia, which allows a smooth untwisting of the lines. This leads to an automatic alignment of the line connection points towards the direction of the power line and thus towards the direction of the wing within the wind window. The required torque for untwisting is realized by the tensile force acting on the power line. The steering lines of the test bench are connected to the ends of the control bar and passed through the center of the

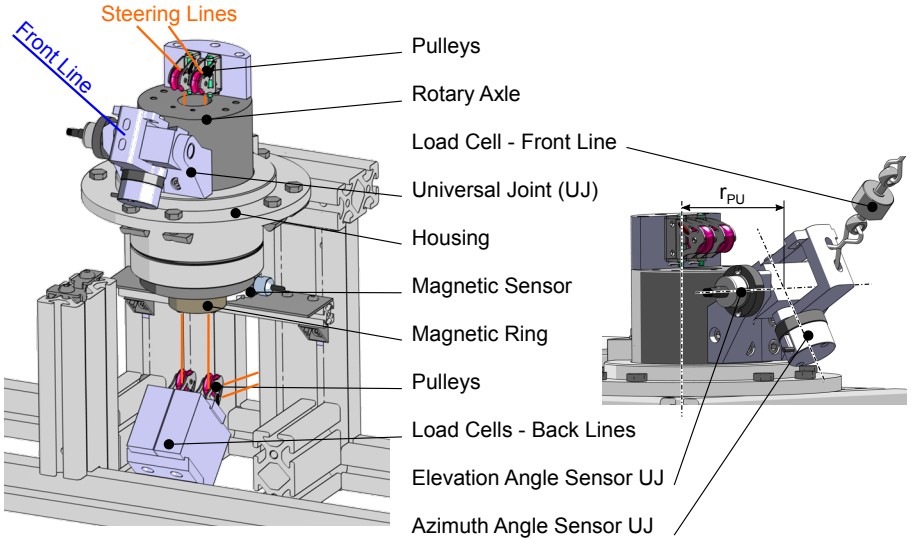

**Figure 7.** Design of the pivot unit

rotary axle to realize minimal inertia. They are redirected by pulleys, connected to rope drums that are operated by motors (see Fig. 5, steering units). The tether forces are measured by means of load cells in the steering lines, not interconnecting the lines. A magnetic sensor attached to the static part measures the rotation of a magnetic ring and thus of the unit itself. The rotary part essentially consists of the rotary axle. The universal joint is attached to it, transmitting the force of the power line.

Each steering unit, which controls the length of a steering line, consists of a cable drum, a gearbox and a motor. The motors are each operated by a servo controller, located within the measurement and control cabinet. The steering units are located in the middle of the test bench, together with the batteries. Since motors and batteries are the heaviest components of the test rig, this arrangement allows the center of gravity to be close to the wheel axis, in order to prevent a static tilting of the trailer (unavoidable tilting of the trailer is measured by an inertial sensor to correct the elevation angle described in Sect. 3.2). The
design force was set to 5000 N. In the front area, in the direction of travel, space was provided for the control cabinets.

### 3.2   Sensor Systems

This section gives a brief overview of the sensor technology, used to achieve the measuring results, which are described in Sect. 5. Components are termed as a sensor system, which serves the purpose of determining certain measuring variables and for which a clear distinction from the overall system is possible. For a complete documentation of all sensor systems please
refer to Hummel (2017).

The exact measurement of the line forces is highly prioritized due to the need for the majority of kite properties (see Sect. 2). To avoid impairments caused by additional masses of the load cells within the steering lines, the load cells are installed without insertion. Furthermore, this also enables the use of load cells with a higher accuracy, which is related to a higher mass of the load cells (HBM S2M, precision class of $0.02\%$, nominal Force $1000N$, which results in an absolute error of $\varepsilon F_{S2M} = \pm0.2N$).

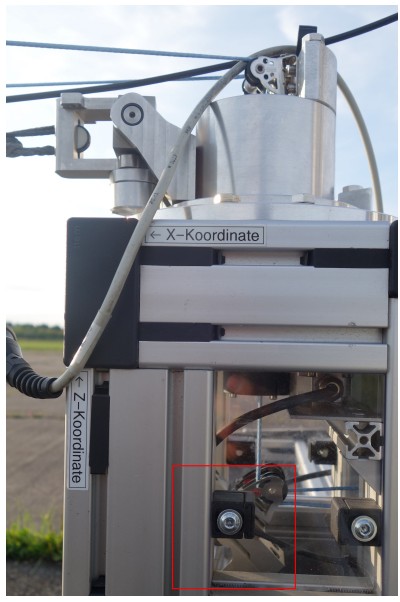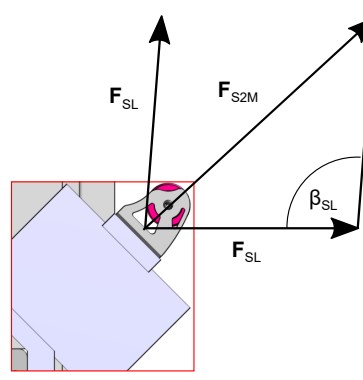

**Figure 8.** The resultant force $F_{res}$ on the load cell of a steering line

The resultant forces $F_{S2M}$ can be obtained from Eq. (8), as illustrated in Fig. 8, assuming that the friction of the pulley is negligible. As shown in Eq. (8), the relation between the force measured at the load cell and the force acting on the steering lines is linear. This is caused by the constant line angle $\beta_{SL}$. With $\beta_{SL} = 90°$ the maximum measurable force within the steering lines is 707 N. Field tests have shown that this value is high enough for common wing sizes. If a higher maximum
force is required in the future, the load cells can be exchanged by sensors with a higher nominal force. However, this will be accompanied by reduced accuracy.

$$F_{S2M} = \sqrt{2 - 2\cos\beta_{SL}}\,F_{SL} \tag{8}$$

The measurement of the force in the power line is performed by an interposition of the load cell (see Fig. 7). A load cell with a nominal force of $5000N$ is used, which has a precision class of $0.2\%$ (HBM U9C). The absolute error results in
$\varepsilon F_{U9C} = \pm 10N$. The signals of the load cells are amplified and then sent to an extension board of the sbRIO. The amplifiers are located as shown in Fig. 5.

Measuring the angle of the power line is intended to enable a simple and reliable determination of the elevation angle $\vartheta$ as well as the azimuth angle $\varphi$, which are illustrated in Fig. 9. The polar coordinate system, and in particular the elevation angle $\vartheta$, is based on Erhard and Strauch (2013a). The definition of the elevation angle is suited for determining the aerodynamic
efficiency, even if the kite is not located within the x-z plane in reference to the wind direction. In contrast to other definitions, i.e. $\beta$ in Schmehl et al. (2013), $\vartheta$ does not vary for a constant glide ratio (see Fig. 9, intersection of red plane with grey wind window). This angle definition facilitates to calculate the glide ratio even if the kite occasionally deflects from the symmetry

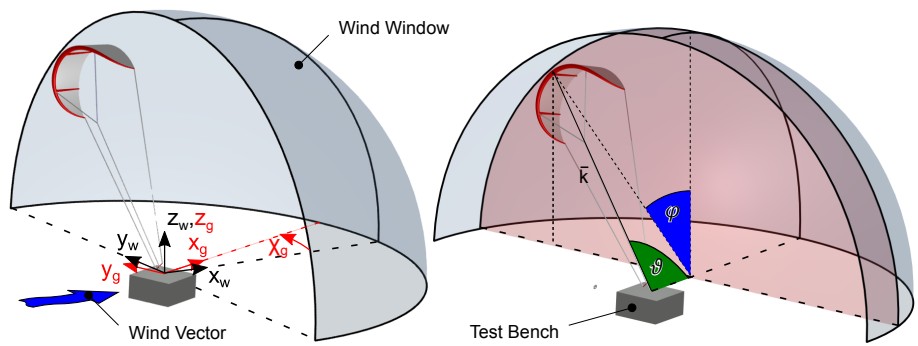

**Figure 9.** left: Cartesian coordinates (for $X_g < 0°$; Index $g$: reference to the test bench; Index $w$: reference to the wind direction); right: elevation angle $\vartheta$ and azimuth angle $\varphi$ (for $X_g = 0°$)

plane of the wind window (downwind position). The rotary axle has a non-neglecting rotational inertia and therefore the measurement of the azimuth and elevation angle, with respect to the test bench, is composed of three sensors, which are shown in Fig. 7. First, the rotational deviation within x-y plane is calculated by the sum of the rotation angle of the rotary axle $\Phi_{RA}$ (measured by the magnetic sensor) and the measured wind direction $X_g$. The magnetic ring of the magnetic sensor has a sufficiently large inner diameter to pass the steering lines through it. Thus, it is possible to mount it underneath the rotary axle without impairing the functionality of the pivot unit. Second, the rotational deviation of the universal joint is measured by the elevation angle sensor ($\Theta_{UJ}$) and the azimuth angle sensor ($\Phi_{UJ}$) to realize low friction as well as a negligible influence on the line angle. As a result, the universal joint will already deflect at low forces in the power line.

The wing position $\mathbf{k_w}$ within the wind window can be calculated by Eq. (11), as a result of the sensors, where index $g$ indicates the reference to the test bench and index $w$ to the wind direction coordinate system.

$$\mathbf{M_w} = \begin{pmatrix} \cos\left(\Phi_{RA}+X_g\right) & -\sin\left(\Phi_{RA}+X_g\right) & 0 \\ \sin\left(\Phi_{RA}+X_g\right) & \cos\left(\Phi_{RA}+X_g\right) & 0 \\ 0 & 0 & 1 \end{pmatrix} \begin{pmatrix} \cos\Theta_{UJ} & 0 & -\sin\Theta_{UJ} \\ 0 & 1 & 0 \\ \sin\Theta_{UJ} & 0 & \cos\Theta_{UJ} \end{pmatrix} \begin{pmatrix} \cos\Phi_{UJ} & -\sin\Phi_{UJ} & 0 \\ \sin\Phi_{UJ} & \cos\Phi_{UJ} & 0 \\ 0 & 0 & 1 \end{pmatrix} \tag{9}$$

$$\mathbf{k_w} = \mathbf{M_w} \begin{pmatrix} r \\ 0 \\ 0 \end{pmatrix} + \begin{pmatrix} \cos\left(\Phi_{RA}+X_g\right) & -\sin\left(\Phi_{RA}+X_g\right) & 0 \\ \sin\left(\Phi_{RA}+X_g\right) & \cos\left(\Phi_{RA}+X_g\right) & 0 \\ 0 & 0 & 1 \end{pmatrix} \begin{pmatrix} r_{PU} \\ 0 \\ 0 \end{pmatrix} \tag{10}$$

$$\mathbf{k_w} = r \begin{pmatrix} \cos\left(\Phi_{RA}+X_g\right)\cos\Theta_{UJ}\cos\Phi_{UJ} - \sin\left(\Phi_{RA}+X_g\right)\sin\Phi_{UJ} \\ \sin\left(\Phi_{RA}+X_g\right)\cos\Theta_{UJ}\cos\Phi_{UJ} + \cos\left(\Phi_{RA}+X_g\right)\sin\Phi_{UJ} \\ \sin\Theta_{UJ}\cos\Phi_{UJ} \end{pmatrix} + r_{PU} \begin{pmatrix} \cos\left(\Phi_{RA}+X_g\right) \\ \sin\left(\Phi_{RA}+X_g\right) \\ 0 \end{pmatrix} \tag{11}$$

With $r$ representing the tether length and $r_{PU}$ representing the distance between the axis of the rotary axle and the pivot point of the universal joint (see Fig. 7). From Eq. (12) the resulting elevation angle $\vartheta_w$ and azimuth angle $\varphi_w$ can be determined.

$$\mathbf{k_w} = r \begin{pmatrix} \cos\vartheta_w \\ \sin\varphi_w \sin\vartheta_w \\ \cos\varphi_w \sin\vartheta_w \end{pmatrix} \tag{12}$$

## 3.3 Error Analysis

The error analysis of the measured data leading to the results of Sect. 5 is described hereafter.

### 3.3.1 Wind Speed

The absolute error of the wind speed measurement for the weather station according to the manufacturer is $\varepsilon v_w = 0.05$ m/s. The error of the wind direction measurement is given by $\varepsilon X = 1°$.

For calculating the kite properties, the resulting wind speed at kite level is needed, whereas the wind speed on top of the 
towing vehicle is measured. Thus, as an additional error for the given test setup, the error due to the height difference in wind measurement must be investigated. The weather station is located on top of the towing vehicle at a height $z_{REF}$ of 3 meters. Depending on the length of the tether, the kite typically reaches a height $z$ of 15 to 30 m. The most commonly used extrapolation method is the wind power law (Akdağ et al., 2013; Ghita et al., 2013). This method is supposed to be valid within the ground level boundary layer ($< 100m$). Empirical data presented by Archer (2013) shows that this model is well suited to 
approximate wind profiles by measuring at a reference height $z_{REF}$ and thus to estimate the wind speed $v_{tw,plaw}(z)$ on kite level $z$. The wind power law is defined as follows,

$$v_{tw,plaw}(z) = v_{tw}(z_{REF}) \left( \frac{z}{z_{REF}} \right)^{\alpha} \tag{13}$$

Here, $v_{tw}(z_{REF})$ indicates the static true wind speed at a fixed position above ground at an altitude $z_{REF}$ (Index $tw$: true wind speed) which also can not directly be measured, because of the moving test bench. The coefficient of friction $\alpha$ depends on 
the terrain type and increases with rising terrain roughness. Despite testing on a former airfield, the coefficient of friction is assessed in an overestimating way, in order to perform a safe calculation (this overestimation will result in an overestimated static wind speed on kite level, which in turn will result in an overestimation of the resulting error $\delta v_{w,real}$). Thus, it is assumed as 0.25 for wooded countryside with many trees. If the true wind vector $v_{tw}(z_{REF})$ points towards the opposite direction of travel, the influence of the relative error $\delta v_{w,real}$ of the wind speed $v_{w,real}(z)$ at kite level will be at a maximum. This is 
because the relative portion of the true wind speed $v_{tw}(z_{REF})$ is maximized and the required speed of the towing vehicle $v_p(z_{REF})$ to reach the desired testing speed $v_w(z_{REF})$ is minimized:

$$v_p(z_{REF}) = v_w(z_{REF}) - v_{tw}(z_{REF}) \tag{14}$$

The resulting wind speed $v_{w,real}$ at flight altitude $z$ is composed of the traveling speed $v_p$ and the theoretical wind speed according to the wind power law $v_{tw,plaw}(z)$:

$$v_{w,real}(z) = v_p(z_{REF}) + v_{tw,plaw}(z) \tag{15}$$

The resulting error is reduced with decreasing altitude, decreasing natural wind and increasing target speed. At present, line lengths of 24 meters are used while the minimum target speed is set to 11 m/s. The relative error can thus be assumed as $\delta v_{w,real} \leq +20\%$. For a detailed calculation please refer to Hummel (2017).

### 3.3.2 Elevation Angle

The angle sensors of the universal joint have an absolute measuring error of $\varepsilon\Theta_{UJ} = \varepsilon\Phi_{UJ} = \pm0.72°$, while the magnetic sensor has an absolute measuring error of $\varepsilon\Phi_{RA} = \pm0.3°$. In order to determine the resultant error from the three angle sensors, the error-prone angles $\vartheta$ and $\varphi$ must be calculated analogously to Sect. 3.2. The maximum error was determined using a MATLAB script. At first, the error-free angles were calculated, followed by a calculation of the error-prone angles for each angle combination. These error-prone angles result from a combination of the minimum and maximum values, which arise due to the individual errors, mentioned before. The maximum error of the elevation angle in the coordinate system of the test bench is $\varepsilon\vartheta_g = 1.2°$. If the error of the weather station $\varepsilon X = 1°$ is added to the error of the magnetic sensor $\varepsilon\Phi_{RA}$, the theoretical maximum error of the elevation angle within the wind direction coordinate system results as $\varepsilon\vartheta_w = 2.1°$.

The quality of the measurement results can be further increased by calculating the line sag and the influence of weight. Nevertheless, as mentioned in Sect. 2, in the scope of this paper post-processing calculations to optimize the estimation of the properties were not carried out, in order to facilitate an easy assessment of the measurement results as well as the reliability of the method.

### 3.4 User Interface

The developed user interface (Barstand) allows to manipulate the control bar position of the test bench. The pilot also receives a haptic feedback of the line forces via the interface. The system was designed based on the assumption that an increase in safety and reliability is achieved by an improved perception of the prevailing flight condition, when a fully or semi manual flight is performed. The pilot should be able to estimate the line forces without numerical display elements in order to extend the pilots perception of the flight situation. As a result, this device allows the subjective evaluation of the kite properties.

The user interface is located inside the towing vehicle and equipped with a common control bar, used to control sport kites (see Fig. 10). The measured line forces are induced to the lines of the user interface by means of winches, operated by servomotors. The force acting on the power line is transfered to the pilot via a harness used for kitesurfing. The motor position and thus the current bar position is determined by integrated encoders. This setup enables a control of the wing, which is close to reality, by moving a common control bar as well as by transmitting the scaled forces acting on the lines. The maximum force of the steering lines was set to 50 N and the force of the power line to 350 N. This determination was made to avoid a

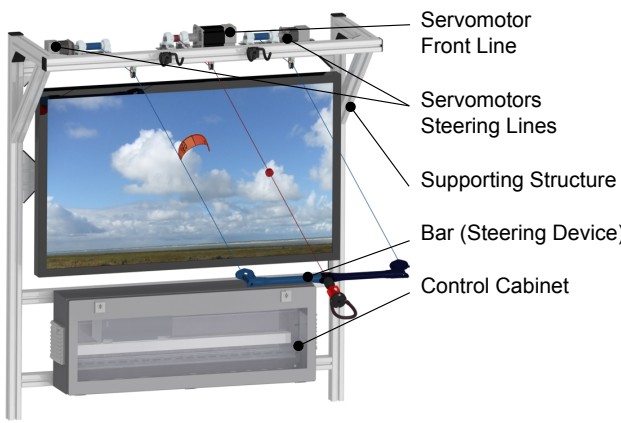

**Figure 10.** Design of the user interface (Barstand)

physical overstressing of the pilot and to limit the size of the actuators. The measured line forces must therefore be scaled by a proportionality factor.

The visual feedback is realized by a display shown in Fig. 10. Caused by the integration of the user interface into the towing vehicle a direct view of the wing is impossible. The image is taken by means of a wide-angle camera atop the roof of the
vehicle. In order to enable a subsequent video evaluation, the recorded data is stored on the camera's internal memory card. When the measurement procedure is started by the pilot, the video recording is initiated automatically by the sbRIO (central control unit, see Sect. 4.1). An LED is placed within the visual range of the camera for a synchronization of the video and the measured data later on. This enables the synchronization of the beginning of data recording with the beginning of the video.

To record the measurement data acquired from the sbRIO, as well as performing control inputs to set up the test run, a
notebook is used as a host computer. The host computer communicates with the sbRIO via network interface. During a test run, the notebook is placed in front of the pilot so that a perception of the numerical display elements of the host computer is possible. During a test procedure the pilot is not required to execute inputs on the host computer.

Furthermore a foot pedal is connected to the host computer which is used to execute maneuvers in the testing mode. When the pedal is actuated by the pilot, the previously set maneuver is executed by the sbRIO. Depending on the degree of automation,
the pilot is enabled to act out certain steering inputs via the control bar. As soon as the pilot releases the pedal, the maneuver is terminated and the kite can be controlled manually again.

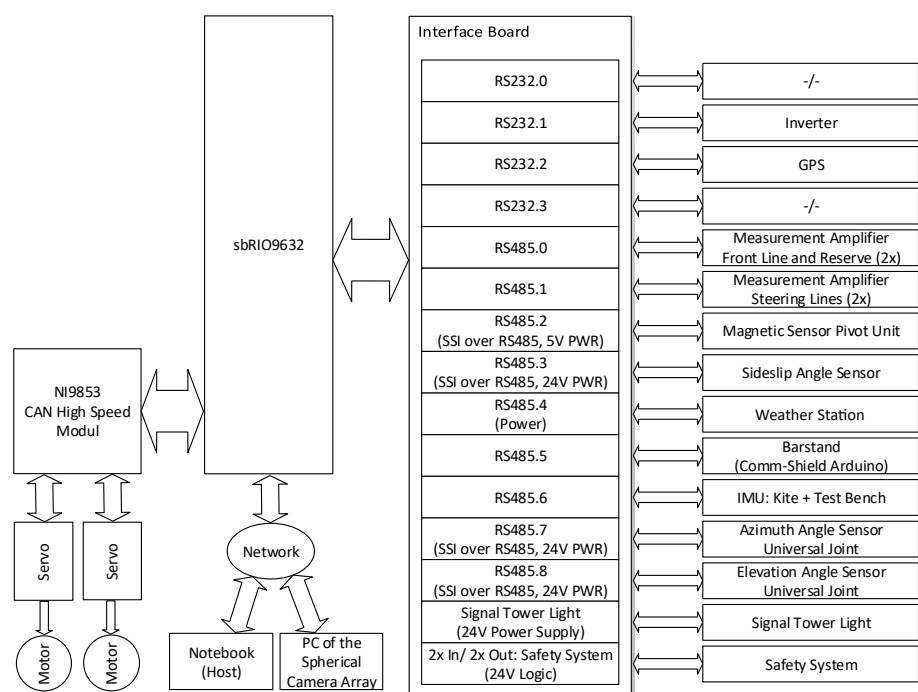

**Figure 11.** Measuring and control diagram

## 4 Data Acquisition

### 4.1 Data Processing System

This section describes the structure of the data processing hardware of the test bench, briefly. A schematic diagram is shown in Fig. 11. The data processing system as well as the DC power supply is localized within the measurement and control cabinet (see Fig. 5). As shown in Fig. 11, the National Instruments' sbRIO 9632 serves as the central control unit. It is connected to various components, such as sensors, via a self made custom interface board. The servo controllers of the motors mentioned in Sect. 3.1 communicate via a CAN module with the sbRIO. A network interface is used to communicate with the host computer as well as retrieving measured values of the spherical camera array. The sbRIO has been chosen because of the implemented central processing unit (CPU) as well as the field programmable gate array (FPGA).

The CPU allows the main control algorithm (the real-time operating system; RTOS) to be executed in real-time. To ensure a safe test operation, a real-time capability has been required. In particular, control inputs have to be executed in a predefined time. For this purpose, a deterministic loop was introduced within the RTOS (with a maximum execution period of 20 ms). This allows the motors to be addressed at a frequency of 50 Hz. The FPGA processor, on the one hand, is used as an access to the analogue and digital interfaces via the internal bus of the sbRIO. On the other hand, programs can be implemented which are

converted into a logic circuit by means of the integrated gates. Due to the configurable logic circuit, a parallel signal processing is possible, which increases the speed of the data processing.

## 4.2  Experimental Setup

The dynamic test procedure, used in this paper, is described below. Dynamic tests are characterized by moving the test bench.
The procedure can be carried out on any straight track. It is of paramount importance that the ground is as plane as possible to reduce oscillation. The measurements within this work have been carried out on the former airport Pütnitz, Germany. The target wind speed was constantly set to 22 kn (11.3 m/s) to demonstrate the repeatability of the test method.

The range of wind speed which could be examined is only limited by the cut-in wind speed of the kite (minimum wind speed for flying the kite) as well as the maximum tensile force resulting from the kite, acting on the test bench (the design force was
set to $5000N$, see Sect. 3.1). Caused by the weight of the test bench the maximum vertical force is currently limited to $3000N$, which could be increased by increasing the weight of the trailer, if necessary. Assuming a coefficient of $C_R = 0.7$ (representing the peak value in Fig. 15), surface area of $A = 10m^2$, air density of $\rho = 1.184kg/m^3$ and apparent wind velocity of $v_a = 50kt$ ($v_a = 25.7m/s$), the resulting Force is $F_R = 2837N < 3000N$ (see Eq. (4)). Since the aerodynamic coefficients investigated so far are wind independent, there is no need to test in higher wind speeds to compare the wings against each other. For the
presented maneuver "Linear Power" in combination with the presented wing sizes, a maximum testing speed of $50kt$ can be given. With crosswind maneuvers an enormous increase in power is expected. If the tensile force exceeds $5000N$ the design of the test bench has to be adapted or the surface area of the wing has to be reduced.

Fig. 12 shows the towing vehicle with the test bench in measuring operation. Measurements are solely conducted on the straight sections. As described above, tests are carried out on days with as little wind as possible. Testing under these conditions
allow a performing of multiple maneuvers without landing the kite since the track can be run both ways.

To launch the kite, it is set up behind the towing vehicle, placed on its trailing edge and lines tightened. When accelerating the test bench, the kite does an ascent movement in the direction of the zenith. The driver of the vehicle is supplied with a display, showing the duplicated view of the host computer. That way, he can assess the current flight situation as well as the currently measured wind speed. The driver adjusts the desired wind speed via the cruise control of the towing vehicle. After
reaching the target speed, the maneuvers can be carried out.

## 4.3  Measurement Data Evaluation

The measurement data is evaluated by means of the software Diadem, which is originated by the company National Instruments, also supplying the software for the host as well as the measurement and control unit.

The implemented script is used to preprocess, process and display the measurement data. First, the desired measurement
files are transferred to the script. Then, each measurement file is preprocessed in a loop. This includes, among other functions, an automatic detection of maneuvers and a distinction between driving along the straight track and turnaround. To obtain the desired graphs, statistical values are calculated from the maneuvers. The graphs as well as an overview of the measured data

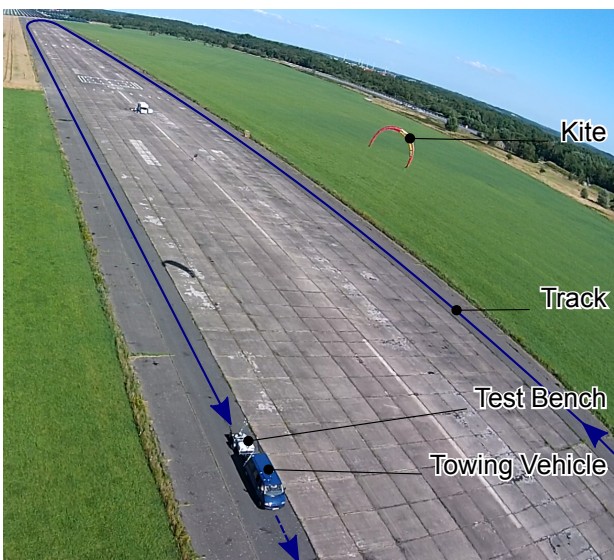

**Figure 12.** Dynamic test procedure

are then added to a report PDF for each measurement file. Once each measurement file has been processed, the results are summarized in an additional overview to allow a comparison between each file.

## 5 Results

This section presents the obtained results for the wing properties defined in Sect. 2. The taken measurements were carried out
by means of the maneuver "Linear Power" in order to demonstrate the functionality of the test bench and the feasibility of the developed test procedure. Before starting the maneuver, the wing is positioned and stabilized by the pilot at the zenith position within the wind window. The foot pedal connected to the host computer is then manually actuated to launch the maneuver. The power position is automatically increased by the sbRIO up to $\Delta l_{PL,max} = 500$ mm (see Fig. 4) with a constant speed over a period of 4.5 seconds. The pilot can still execute steering inputs to keep the kite in a stable position at the zenith.

The measurement diagrams are shown in the following subsections. Only maneuvers lasting a given minimal timespan were taken into account. During some maneuvers an unintentional change of position or orientation (e.g. caused by gusts) led to the pilot aborting the maneuver, which can result in a too short maneuver, which in turn would make the statistical value calculation impossible. The valid results are plotted against the power ratio $u_p$. The determination of the angle of attack was not feasible within the scope of this work and will be done in future research of this project.

### 5.1 Tested Kites

For characterization of the dynamic properties, five different kites with the same surface area of $10 \text{ m}^2$ were measured (denoted with kite A to E within the graphs). All kites are designed for different purposes in kite sport:

On the one hand, kite C was designed to ride efficiently upwind, i.e. affording a high traveling angle in wind direction. In addition, high jumps with a long air time should be possible. Therefore, a high aerodynamic efficiency associated with a high resulting force is required. Furthermore this kite should provide a high depower capability, resulting in a significant change of the lift coefficient.

The kites D and E have the same design, but originated from different model years. Because of their shape, these kites feature a significant contrast to the other kites. Significantly more wing area is located at the wing tips, which should result in lower aerodynamic efficiency as well as a lower lift coefficient.

The Kite A is supposed to be an all-rounder which means the resulting lift and efficiency should be positioned between C and D/E.

Kite B is designed to achieve good handling and turning abilities as well as providing a good upwind performance at the same time. For this reason the steering forces have to be higher while depowered ($u_p \simeq 0$), compared to the other kites.

The measurements were conducted during two different days (marked with day 1 and 2). For each property, a figure is shown which summarizes all measurement data into a single curve for each kite in order to compare the kites against each other. Additionally, these figures show the resulting error resulting from all maneuvers which where taken into account for a

confidence interval of $95\%$.

## 5.2    Aerodynamic Efficiency

The measurement results of the elevation angle $\vartheta_w$ can be seen in Fig. 13. The resulting aerodynamic efficiency can be calculated by Eq. (6) (see Fig. 14). The different curves can be distinguished by height and progression.

As discussed in the previous chapter, it can be shown that kite C offers the highest and kites D/E the lowest aerodynamic

efficiency. It can also be concluded, that a reliable repeatability within the same day is given. This finding was confirmed by further tests on different days. The only significant deviation was found after a long period between two test runs. The time between day 1 and day 2 was exactly one year. The elevation angle differs between these days only by an offset. In order to determine this offset in the future and, if necessary, to compensate it, a reference kite was introduced, which is measured once every test day. The resulting curves of this reference kite should fit each other on different test runs. If an offset occurs, the

starting points of the graphs can be corrected and thus the wings can still be compared relatively to each other. In order to fully compensate this deviation in the future, the initial horizontal alignment of the test bench will be measured by means of an inertial measurement unit. The deviation is most likely caused by changes in geometry being difficult to control, for example, a change in tire pressures of the trailer or the towing vehicle.

## 5.3    Lift Coefficient

The lift coefficient $C_L$ is calculated according to Eq. (2) using the given manufacturer's surface area of 10 m$^2$ and a constant air density of $\rho = 1.184 \frac{kg}{m^3}$. The airflow velocity is assumed to equal the measured wind speed of the weather station. The total tether force is calculated by the sum of the measured forces of three load cells. Due to the high elevation angles, the resulting force coefficient $C_R$ resembles $C_L$ and is not shown separately (see Eq. (4) and Eq. (2) with $\sin(\vartheta_w > 70°) \approx 1$)

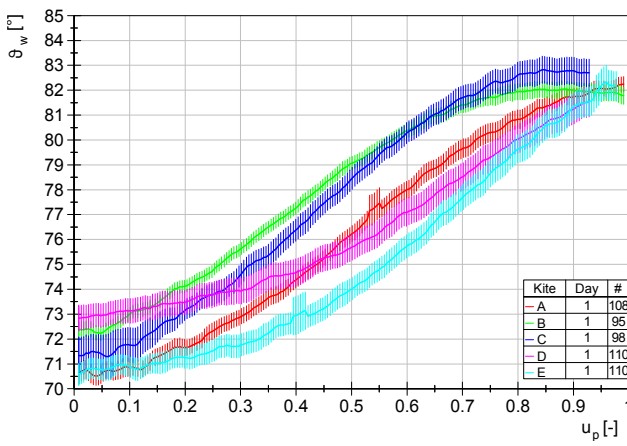

**Figure 13.** Elevation angle with resulting error ($P = 95\%$)

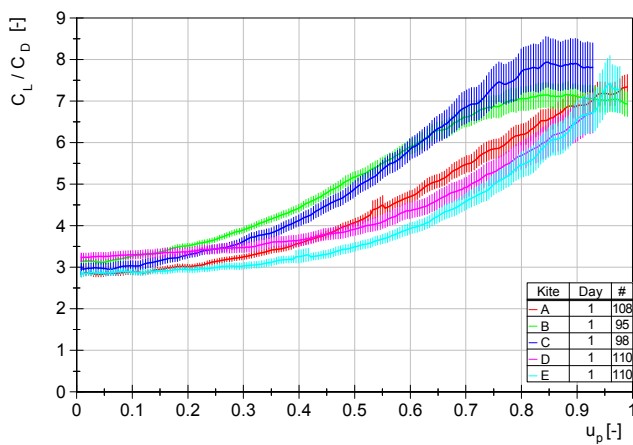

**Figure 14.** Aerodynamic efficiency (lift-to-drag ratio) with resulting error ($P = 95\%$)

The resulting curves of the data sets are shown in Fig. 15. As predicted in Sect. 5.1, kite C offers the highest and kites D/E the lowest lift coefficient. The deviation between datasets of the same kite lies within the resulting error. The influence of the above-mentioned deviation of the elevation angle measurement on the lift coefficient is negligible.

As mentioned in Sect. 2.1, the depower capability for each kite can be calculated by the difference between the maximum and minimum values. Apparently, kites B and C are best suited for AWE systems using the pumping mode, because of their high depower capability as well as their high lift coefficient. A further distinction can be made based on the curve progressions. Kites A to C can be characterized by their degressive progression, whereas kites D and E are characterized by a progressive increase of the lift coefficient.

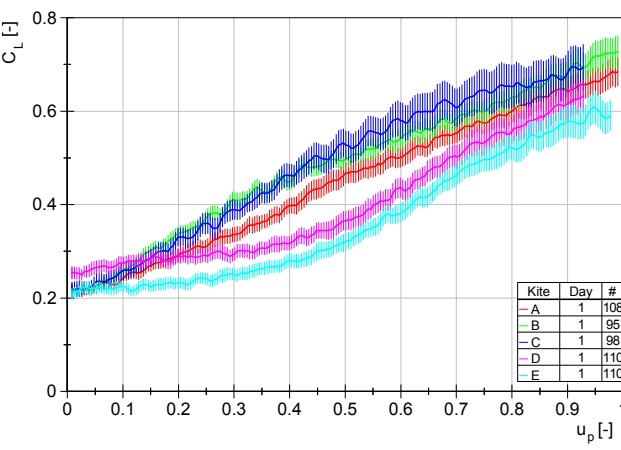

**Figure 15.** Lift Coefficient with resulting error ($P = 95\%$)

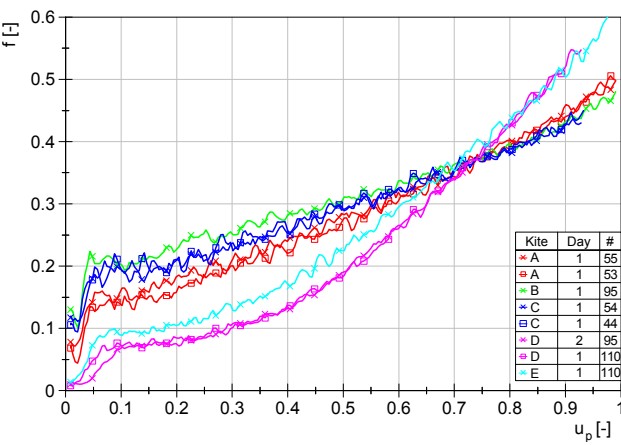

**Figure 16.** All measurement files: Force ratio between steering lines and power line

## 5.4 Force ratio

Figures 16 and 17 show the force ratio $f$ between steering lines and power line which can be calculated by Eq. (7). In order to be able to estimate the reproducibility, for each kite property all eight data sets are first presented together within the same diagram (Fig. 16). Obviously, a distinction between the kites is possible. As a result of their different wing shapes, the curve
5   progression of the kites D and E compared to the other kites is clearly different (progressive). Furthermore, the kites can be distinguished by height of the force ratio. With these curves as well as the force curves itself existing simulation models can be evaluated reliably.

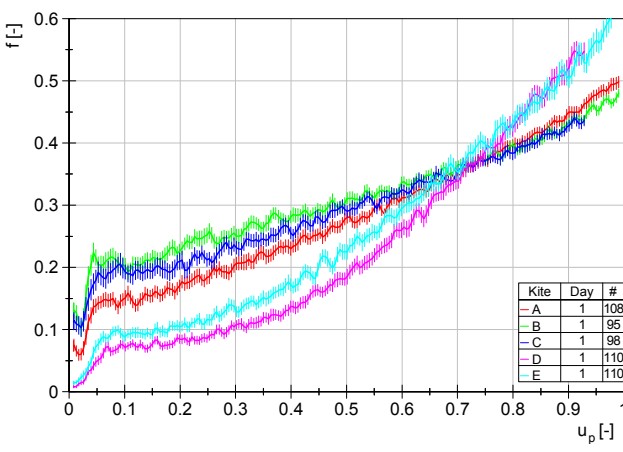

**Figure 17.** Force ratio between steering lines and power line with resulting error ($P = 95\%$)

For AWE systems the force ratio is of great importance, since it determines the steering possibility of the wing while fully depowered (especially during retraction phase). To guarantee the execution of control commands by transmitting the steering forces, the force ratio must not be too low.

## 6    Conclusion and Outlook

In most cases, a reproducible, high-quality measurement of flight dynamic properties of tethered flexible membrane wings exceeds the available budget. Furthermore, existing approaches do not allow for a recording or even automation of steering inputs, which is crucial for the reproducibility of the experiment. In this paper, we have presented a unique tow test setup for automatic measurement of the dynamic properties of different wing types at full scale and under realistic conditions. The objective was to demonstrate the methodology and particularly the repeatability of the test procedure. Using the maneuver "Linear Power", we determine the aerodynamic coefficients and lift-to-drag ratio of the wing as functions of the ratio of power and steering line lengths – denoted as relative power setting – by measuring line forces and line angles. The ratio is varied automatically, while the pilot is manually adjusting the steering line lengths to keep the kite at a fixed position relative to the towing vehicle. By automating the test cycles we can acquire mean values of high statistical quality with minimal errors. We have demonstrated the repeatability on the basis of eight recorded data sets using the maneuver "Linear Power" at a constant wind speed of 22 kn (11.3 m/s). We conclude from this study that it is feasible to objectively measure the flight dynamic properties of tethered membrane wings and to quantitatively assess and compare different wing designs.

Based on this work, we propose several functional enhancements for future research. By performing more sophisticated flight maneuvers the full operational envelope of AWE systems can be covered. By completing the automation of the process we expect a significant increase of measurement accuracy which will improve future aerodynamic parameter identification as

well as evaluation of existing simulation models. A further accuracy increase can be achieved by adding sensors to the wing measuring directly the flight state and the relative flow.

*Code availability.* The code and measurement data can be made available in the framework of a cooperation agreement. If interested please contact the corresponding author.

5    *Competing interests.* The authors declare that they have no conflict of interest.

*Acknowledgements.* Roland Schmehl has received funding from the European Union's Horizon 2020 research and innovation programme under the Marie Skłodowska-Curie grant agreement No. 642682 for the ITN project AWESCO and the grant agreement No. 691173 for the "Fast Track to Innovation" project REACH.

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
