# Peer review of "Automatic Measurement and Characterization of the Dynamic Properties of Tethered Membrane Wings"

_Wind Energy Science, 2018_

## Referee Comment (RC1) · J. Deparday (Referee) · 7 Sep 2018

**1   general comments**

The article presents an original and well-designed experimental setup to assess the flight dynamic properties of kites used for airborne wind energy system. This full-scale experimental setup shows great promises for reliable and repeatable experiments of general performance and dynamic maneuvers.
I found this paper well written with clear figures to explain this complex test bench and the first results. I was sometimes a bit confused with the structure of the article but the main story can easily be followed.

[Figure]

**2 specific comments**

To improve the overall good quality of this paper, I would have a few remarks:

**2.1 terms in introduction part**

In the introduction, there are many specific terms specific to kite/flexible membrane of airborne wind energy system (ex: Leading-Edge Inflatable tube kites, ram air wing, chambered design, ribs, ...) As the readers of this journal might not be completely aware of all these terms, either explaining a bit more the most important terms or an extra figure describing the general type of kite used with this test bench might help to visualize the topic. A photograph or a figure inspired from Figure 2 could be an idea.

**2.2 control of the control bar**

I don't really understand how the control bar is controlled and slides along the power line.

**2.3 load cell precision**

The load cells are presented in section 3.2 with the dimensionless precision. Their absolute error in Newton are explained a few pages after. It would be more convenient to briefly insert these absolute errors in section 3.2.

**2.4   wind speed error**

I don't really understand the paragraph page 12 from line 16 to 23. What is your point in this explanation? Why is it safe to overestimate the roughness?

**2.5   elevation angle**

As you know the measuring error of the angle sensors, and you have the formulas to determine the azimuth angle, did you use a GUM ( Guide to the expression of uncertainty in measurement) analysis?

**2.6   data processing system**

It is written: "A network interface is used to communicate with the host computer as well as retrieving measured values of the spherical camera array". What are the measured values of the spherical camera array? The images?

**2.7   synchronization of the measurements**

Do you record data "on the flow" i.e. sensors send raw data as fast as they can, or does the sbRIO control them using a "master-slave" communication? What are the sampling frequencies of the sensors? As this test bench will be used for dynamic measurements, the dynamics of the sensors and of the recording might be relevant to add in the article.

**2.8   results presentation**

It is written: "Only maneuvers lasting a given minimal timespan were taken into account. During some maneuvers an unintentional change of position or orientation (e.g. caused by gusts) led to the pilot aborting the maneuver. The valid results are plotted..."
What is the minimal timespan?
To clearly understand: the maneuvers affected by gusts are not part of the valid results?

**2.9   presentation of the tested kites**

It is written: "a figure is shown which summarizes all measurement data into a single curve... Additionally, these figures show the resulting error for a confidence interval of 95%".
Does this single curve is the average of all the tests for each kite?
Is the confidence interval of 95% coming from the deviation between the different tests, or from the uncertainty measured in section 3.3.3?

**2.10   discussion on the elevation angle offset**

It is written: "The only significant deviation was found after a long period between two test runs".
Are these 2 test runs described here are the same than the 2 different days (day 1 and day 2) presented in the figures?

I don't really understand what a "reference wing" could be.
More as a suggestion: would it be possible to rotate "manually" the 3 rotations axes (universal joint and rotary axle)? By recording independently the angles purposely set,

a calibration of the 3 angle sensors might be able to calibrate and check any potential offset?

**2.11   Results of lift coefficient**

It is written: "The airflow velocity is assumed to equal the measured wind speed of the weather station." For these results, do you neglect the wind power law equation (equation 13)?

**2.12   force ratio**

Are the angles of the steering lines relative to the power line taken into account? Or as the kite is high enough, are they considered parallel to the power line?

**2.13   Conclusions**

It is written: "So far the reproducible measurement of flight dynamic properties of tethered flexible wings was not feasible".
Was it not feasible?  Or not done yet?  It sounds in contradiction with the abstract: "We can conclude from this study that an objective measurement of specific dynamic properties of highly flexible membrane wings is feasible".

**3   technical corrections**

I did not detect any typing errors or wrong sentences.  There is only one small typing error in figure 10: "Wheater station"

---

## Referee Comment (RC2) · Anonymous Referee #2 · 22 Sep 2018

**1. Summary and Recommendation**

The manuscript describes an automated test bench setup for measurements of flight properties of tethered wings. The system is build on a car trailer to be used in towing mode and features extended steering and measurement components. After an introduction, the quantities to be measured are introduced. Subsequently, the hardware setup is described in detail. After a brief section on data acquisition and testing procedure, flight test results are presented by comparing aerodynamic properties of five kites in static flight mode.

In general and in large part, the paper is well written, contains clear figures and provides detailed insights in the technical implementation of the setup. As these measurements are essential for the development of kites for airborne wind energy (AWE),

the manuscript is of broad interest for the AWE community. Thus, this clear and consistent presentation of the experimental setup can be clearly regarded as the main contribution of the paper. In contrast, the result section appears weak as only static flight at one wind speed is discussed which by far does not cover the operational range needed in AWE setups. However, rating the overall manuscript, the reviewer definitely recommends publication in WES. The discussion of the data and especially the outlook should undergo a (minor) rework in order to provide a clearer assessment of the experimental results achieved and future experiments needed to provide a full characterization of tethered wings for application in AWE. Please find details below.

2. Scientific questions and issues

- One big advantage of the setup is that the wind speed can be directly adjusted by just setting the cruising speed of the towing vehicle. Why are only results for one wind speed presented. Could you specify and discuss range of wind speeds which could be examined by this setup?

- AWE setups require a dynamic flight mode of the kites. How will dynamic flight test be implemented in the existing setup?

- The Abstract should be shortened. In parts, it resembles an introduction but should provide a condensed summary of the own work presented in the paper.

- The "Conclusion and Outlook" section has a partly confusing structure and should be reworked. In the first two sentences, it is stated, that "dynamic flight...was not feasible...is essential". Two sentences later, the authors claim that "...presented work fills this gap...". Subsequently a lot of issues are addressed but in arbitrary order in one long paragraph. Please state clearly what has been achieved. Then it would be nice to have a summary of future work to be required by AWE applications and a brief discussion of the ideas to extend the setup.

- the line sag is mentioned in the outlook, but shouldn't it be discussed in the error

analysis (3.3.3), especially for static depowered flight?

- are the errorbars in Figure 14 realistic as the C_L coefficient depends on wind speed, for which an error of 20% is assumed (3.3.2)?

3. Technical corrections

- consistent symbols should be used, e.g. for vectors (bold face on page 4 <-> overline on page 11)

---

## Author Response (AR1)

**1 general comments**

The article presents an original and well-designed experimental setup to assess the flight dynamic properties of kites used for airborne wind energy system. This full-scale experimental setup shows great promises for reliable and repeatable experiments of general performance and dynamic maneuvers.
I found this paper well written with clear figures to explain this complex test bench and the first results. I was sometimes a bit confused with the structure of the article but the main story can easily be followed.

[Figure]

**2 specific comments**

To improve the overall good quality of this paper, I would have a few remarks:

**2.1 terms in introduction part**

In the introduction, there are many specific terms specific to kite/flexible membrane of airborne wind energy system (ex: Leading-Edge Inflatable tube kites, ram air wing, chambered design, ribs, ...) As the readers of this journal might not be completely aware of all these terms, either explaining a bit more the most important terms or an extra figure describing the general type of kite used with this test bench might help to visualize the topic. A photograph or a figure inspired from Figure 2 could be an idea.

**2.2 control of the control bar**

I don't really understand how the control bar is controlled and slides along the power line.

**2.3 load cell precision**

The load cells are presented in section 3.2 with the dimensionless precision. Their absolute error in Newton are explained a few pages after. It would be more convenient to briefly insert these absolute errors in section 3.2.

[Figure]

**2.4   wind speed error**

I don't really understand the paragraph page 12 from line 16 to 23. What is your point in this explanation? Why is it safe to overestimate the roughness?

**2.5   elevation angle**

As you know the measuring error of the angle sensors, and you have the formulas to determine the azimuth angle, did you use a GUM ( Guide to the expression of uncertainty in measurement) analysis?

**2.6   data processing system**

It is written: "A network interface is used to communicate with the host computer as well as retrieving measured values of the spherical camera array". What are the measured values of the spherical camera array? The images?

**2.7   synchronization of the measurements**

Do you record data "on the flow" i.e. sensors send raw data as fast as they can, or does the sbRIO control them using a "master-slave" communication? What are the sampling frequencies of the sensors? As this test bench will be used for dynamic measurements, the dynamics of the sensors and of the recording might be relevant to add in the article.

[Figure]

**2.8 results presentation**

It is written: "Only maneuvers lasting a given minimal timespan were taken into account. During some maneuvers an unintentional change of position or orientation (e.g. caused by gusts) led to the pilot aborting the maneuver. The valid results are plotted..."
What is the minimal timespan?
To clearly understand: the maneuvers affected by gusts are not part of the valid results?

**2.9 presentation of the tested kites**

It is written: "a figure is shown which summarizes all measurement data into a single curve... Additionally, these figures show the resulting error for a confidence interval of 95%".
Does this single curve is the average of all the tests for each kite?
Is the confidence interval of 95% coming from the deviation between the different tests, or from the uncertainty measured in section 3.3.3?

**2.10 discussion on the elevation angle offset**

It is written: "The only significant deviation was found after a long period between two test runs".
Are these 2 test runs described here are the same than the 2 different days (day 1 and day 2) presented in the figures?

I don't really understand what a "reference wing" could be.
More as a suggestion: would it be possible to rotate "manually" the 3 rotations axes (universal joint and rotary axle)? By recording independently the angles purposely set,

a calibration of the 3 angle sensors might be able to calibrate and check any potential offset?

**2.11 Results of lift coefficient**

It is written: "The airflow velocity is assumed to equal the measured wind speed of the weather station." For these results, do you neglect the wind power law equation (equation 13)?

**2.12 force ratio**

Are the angles of the steering lines relative to the power line taken into account? Or as the kite is high enough, are they considered parallel to the power line?

**2.13 Conclusions**

It is written: "So far the reproducible measurement of flight dynamic properties of tethered flexible wings was not feasible".
Was it not feasible? Or not done yet? It sounds in contradiction with the abstract: "We can conclude from this study that an objective measurement of specific dynamic properties of highly flexible membrane wings is feasible".

**3  technical corrections**

I did not detect any typing errors or wrong sentences. There is only one small typing error in figure 10: "Wheater station"

[Figure]

[Figure]

Wind Energ. Sci. Discuss.,
https://doi.org/10.5194/wes-2018-56-AC1, 2018

[Figure]

Thank you for your detailed review. I will answer your questions in the same order:

**2.1 terms in introduction part**

I will explain the kite-specific terms within the paper, briefly.

i.e. "ribs (connecting topskin and bottomskin)"

"chambered design (chambers are separated by ribs, topskin and bottomskin)"

Additionally, we can add a figure, as shown below (Fig.1)

[Figure]

Caption: "left: Leading Edge Inflatable (LEI) tube kites (single-skin kite); right: ram air wing"

Alternatively, the terms are also explained in more detail in the following publications:

**Ram air wing:**

Dunker, Storm (2014): Ram-air Wing Design Considerations for Airborne Wind Energy. In: Uwe Ahrens, Moritz Diehl und Roland Schmehl (Hg.): Airborne wind energy: Springer-Verlag (Green energy and technology), S. 517–546. http://dx.doi.org/10.1007/978-3-642-39965-7_31.

p.518 Fig. 31.1

**LEI kite**

Bosch, Allert; Schmehl, Roland; Tiso, Paolo; Rixen, Daniel (2014): Nonlinear Aeroelasticity, Flight Dynamics and Control of a Flexible Membrane Traction Kite. In: Uwe Ahrens, Moritz Diehl und Roland Schmehl (Hg.): Airborne wind energy: Springer-Verlag (Green energy and technology), S. 307–323. http://dx.doi.org/10.1007/978-3-642-39965-7_17.

p.312 Fig. 17.3

What do you think is the most suitable solution?

**2.2 control of the control bar**

The steering lines of the test bench are connected to the ends of the control bar, passed through the rotary axle and redirected by pulleys. At the other side, they are connected to rope pulleys that are operated by motors (p.8 lines 14-15, Fig. 6).

[Figure]

The sliding of the control bar along the power line is shown in Fig. 3. In the center of a common kitesurfing control bar, there is a hole through which the powerline is guided.

**2.3 load cell precision**

We can insert the absolute error in section 3.2. and delete chapter 3.3.1

**2.4 wind speed error**

Actually, we are measuring the wind speed on top of the towing vehicle, while driving. However, for calculating the kite properties, the resulting wind speed at kite level is needed. The resulting error caused by differences in height can be estimated by the wind power law (eq. 13).

An overestimated coefficient of friction will result in an overestimated static wind speed on kite level, which in turn will result in an overestimation of the resulting error $\delta v_{w,real}$

Here is an example:

We have to adjust the traveling speed of the towing vehicle depending on the measured wind speed on the towing vehicle which in turn has to be the target testing speed (i.e. $v_w(z_{REF}) = 11m/s$). As can be seen in eq. 14, the relative portion of the traveling speed depends on the natural wind speed at the height of $z_{REF} = 3m$. The traveling speed of the towing vehicle is minimized if the vector of the static wind speed is directed against the towing direction. Assuming a maximum line length of 24 meters ($z = 24m$) as well as a maximum natural wind speed of 3m/s at a height of 3 meters ($v_{tw}(z_{REF}) = 3m/s$) with eq. 13. a static wind speed on kite level ($z = 24m$) of $v_{tw,plaw}(z) = 5m/s$ results.

[Figure]

By a target speed $v_w(z_{REF})$ of $11m/s$, a traveling speed of $v_p = 8m/s$ results (eq. 14).

The wind speed on kite level is composed of the traveling speed $(v_p(z_{REF}))$ as well as the wind speed calculated with the wind power law (eq. 15). Since the traveling speed of the wind window is constant $(v_p(z_{REF} = 3m) = v_p(z_{REF} = 24m))$, the true wind speed on kite level is calculated to $v_{w,real}(z = 24m) = 13m/s$ if $v_w(z_{REF}) = 11m/s$.

Regarding this worst-case scenario the relative error can be assumed as $\delta v_{w,real} <= +20\%$.

**2.5 elevation angle**

The resultant error from the three angle sensors and the accuracy of the weather station was calculated analogously to the elevation angle (Sec.3.3.3) with $\epsilon\varphi_w = 2.14°$. However, this angle is not used for the calculation of the dynamic properties and was therefore not mentioned in this paper.

**2.6 data processing system**

The camera system works as an independent sensor system and also provides the elevation angle as well as the azimuth angle. In addition, the orientation is measured to improve future control algorithms. For the calculation of the dynamic coefficients it has not yet been used.

**2.7 synchronization of the measurements**

all presented sensors are operated via the "master-slave" principle. Within the deterministic loop mentioned in Sec. 4.1 a measured value is retrieved once per

iteration (50Hz), except for the weather station, which has a frequency of 1Hz. The measuring amplifiers of the load cells could theoretically reach a frequency of 150Hz and the angle sensors of approx. 100Hz.

**2.8 results presentation**

The minimal timespan in postprocessing was set to 4 seconds.

Your question was: "the maneuvers affected by gusts are not part of the valid results?"

This is not completely right, only maneuvers with a length less than the mentioned 4 seconds were excluded. This means if the pilot was able to control the kite until the end of the maneuver, also maneuvers which were strongly influenced by gusts were taken into account.

**2.9 presentation of the tested kites**

Yes, this curve represents the average of all maneuvers for each kite (including different test runs on the same day). The confidence interval results from the deviation of the different maneuvers.

**2.10 discussion on the elevation angle offset**

Since the curves of day 1 and day 2 only differ by an offset for the aerodynamic efficiency, a representation was omitted to improve the clarity. Of course, this offset must be further investigated if it cannot be compensated by the measures mentioned below (p.17 line 32 f.).

[Figure]

For the force ratio, each test run is shown individually as an example. An offset was not observed. Regarding your suggestion of calibration: We carried out this kind of calibration. The offset is very unlikely caused by the angle sensors. As mentioned, the offset is most likely caused by changes in geometry which are difficult to control. This is why we will also test the reference wing/ kite in future.

We test this reference kite each test day. The curves of different days should fit each other. If this is not the case, the setup changes compared to the test days before must be investigated systematically. Furthermore, the offset can be compensated in postprocessing, so that the kites can be still compared against each other. As mentioned before, it can also be determined from other measurements that curves for a kite fit very well against each other at the same day. The assumption that this offset is caused in changes of geometry, e.g. due to the changed tire pressure or loading of the towing vehicle, is therefore very likely.

**2.11 Results of lift coefficient**

No, we do not neglect the wind power law. Here the error of the wind speed, discussed before is important. If we have perfect wind conditions, which means zero wind speed $(v_{tw}(z_{REF}) = 0m/s)$, the wind speed resulting from the wind power law is zero, too, which in turn means the resulting windspeed on kite level is the same as on the ground. If the natural wind is not equal to zero (which is always the case), the relative error is below +20%.

**2.12 force ratio**

The angles of the steering lines are not taken into account because of the line length.

[Figure]

**2.13 Conclusions**

This is correct, the wording is not favorable. We will change it to:

'With existing approaches the reproducible measurement of flight dynamic properties of tethered flexible wings was not feasible."

[Figure]

[Figure]

Struts
Canopy
Leading Edge
Trailing Edge
Topskin
Bottomskin
Ribs

**Fig. 1.**

[Figure]

Wind Energ. Sci. Discuss.,
https://doi.org/10.5194/wes-2018-56-RC2, 2018

[Figure]

The manuscript describes an automated test bench setup for measurements of flight properties of tethered wings. The system is build on a car trailer to be used in towing mode and features extended steering and measurement components. After an introduction, the quantities to be measured are introduced. Subsequently, the hardware setup is described in detail. After a brief section on data acquisition and testing procedure, flight test results are presented by comparing aerodynamic properties of five kites in static flight mode.

In general and in large part, the paper is well written, contains clear figures and provides detailed insights in the technical implementation of the setup. As these measurements are essential for the development of kites for airborne wind energy (AWE),

the manuscript is of broad interest for the AWE community. Thus, this clear and consistent presentation of the experimental setup can be clearly regarded as the main contribution of the paper. In contrast, the result section appears weak as only static flight at one wind speed is discussed which by far does not cover the operational range needed in AWE setups. However, rating the overall manuscript, the reviewer definitely recommends publication in WES. The discussion of the data and especially the outlook should undergo a (minor) rework in order to provide a clearer assessment of the experimental results achieved and future experiments needed to provide a full characterization of tethered wings for application in AWE. Please find details below.

2. Scientific questions and issues

- One big advantage of the setup is that the wind speed can be directly adjusted by just setting the cruising speed of the towing vehicle. Why are only results for one wind speed presented. Could you specify and discuss range of wind speeds which could be examined by this setup?

- AWE setups require a dynamic flight mode of the kites. How will dynamic flight test be implemented in the existing setup?

- The Abstract should be shortened. In parts, it resembles an introduction but should provide a condensed summary of the own work presented in the paper.

- The "Conclusion and Outlook" section has a partly confusing structure and should be reworked. In the first two sentences, it is stated, that "dynamic flight...was not feasible...is essential". Two sentences later, the authors claim that "...presented work fills this gap...". Subsequently a lot of issues are addressed but in arbitrary order in one long paragraph. Please state clearly what has been achieved. Then it would be nice to have a summary of future work to be required by AWE applications and a brief discussion of the ideas to extend the setup.

- the line sag is mentioned in the outlook, but shouldn't it be discussed in the error

analysis (3.3.3), especially for static depowered flight?

- are the errorbars in Figure 14 realistic as the C_L coefficient depends on wind speed, for which an error of 20% is assumed (3.3.2)?

3. Technical corrections

- consistent symbols should be used, e.g. for vectors (bold face on page 4 <-> overline on page 11)
* * *
[Figure]

Wind Energ. Sci. Discuss.,
https://doi.org/10.5194/wes-2018-56-AC2, 2018

[Figure]

Thank you for your review. In principle, your comments regarding the experimental data are justified. The scope of this work was to demonstrate the proper functioning of the test bench, specifically the repeatability of the test procedure. Indeed, to fully characterize flexible membrane wings, more sophisticated maneuvers are needed. Due to the time constraints it was not possible to implement further maneuvers within the scope of this work. At the moment, however, the department is working on exactly this functional enhancement and on a fully automation. In the abstract we therefore mention: "On the basis of this work, more complex maneuvers and a full degree of automation can be implemented in subsequent work. It can also be used for aerodynamic parameter identification. "

[Figure]

And within conclusion and outlook: "in order to increase reproducibility and perform aerodynamic parameter identifications, it is necessary to implement and automate more sophisticated maneuvers"

**2 Scientific questions and issues**

**Referee:** *"One big advantage of the setup is that the wind speed can be directly adjusted by just setting the cruising speed of the towing vehicle. Why are only results for one wind speed presented. Could you specify and discuss range of wind speeds which could be examined by this setup?"*

The main reason for presenting data recorded at the same wind speed was to demonstrate the repeatability. Otherwise, from the authors point of view, an assessment of repeatability is not feasible.

The range of wind speed which could be examined is only limited by the cut-in wind speed of the kite (minimum wind speed for flying the kite) as well as the maximum tensile forces resulting from the kite, acting on the test bench (the design force was set to $5000N$, which is described in Sec.3.1). Caused by the weight of the test bench the maximum vertical force is actually limited to $3000N$, which could be increased by increasing the weight of the trailer, if necessary.

Assuming a force coefficient of $C_R = 0.7$ (which is the maximum value in Fig. 14), surface area of $A = 10m^2$, air density of $\rho = 1.184kg/m^3$ and apparent wind velocity of $v_a = 50kt$ $(v_a = 25.7m/s)$, the resulting Force is $F_R = 2837N < 3000N$ (see Eq.4). But since the aerodynamic coefficients investigated so far should be wind independent, the authors see no need to test in higher wind speeds to compare the wings against each other.

As can be seen from Eq.4, the tensile force mainly depends on the wing size and the

[Figure]

Interactive
comment

apparent wind velocity. For the presented maneuver "Linear Power" and the presented wing sizes, a maximum testing speed of $50 kt$ can be given, which can be increased, if necessary.

With crosswind maneuvers an enormous increase in power is expected. If the tensile force exceeds $5000 N$ the design of the test bench has to be adapted or the wing has to be smaller.

**Referee:** *"AWE setups require a dynamic flight mode of the kites. How will dynamic flight test be implemented in the existing setup?"*

Dynamic flight maneuvers are currently being implemented and have (partialy) already been successfully tested, but this exceeds the scope of this paper. Automated maneuvers can be implemented by newly developed control algorithms with the help of the presented sensors as well as additionally developed sensors.

**Referee:** *"The Abstract should be shortened. In parts, it resembles an introduction but should provide a condensed summary of the own work presented in the paper."*

We will shorten the abstract and move some content to the introduction part.

**Referee:** *"The 'Conclusion and Outlook' section has a partly confusing structure and should be reworked. In the first two sentences, it is stated, that 'dynamic flight...was not feasible...is essential'. Two sentences later, the authors claim that '...presented work fills this gap...'. Subsequently a lot of issues are addressed but in arbitrary order in one long paragraph. Please state clearly what has been achieved. Then it would be nice to have a summary of future work to be required by AWE applications and a brief discussion of the ideas to extend the setup."*

We will revise the section "Conclusion and Outlook" for a more clear structure.

[Figure]

Regarding future work, a rough outlook can be given. When writing this paper the recommendation to subsequent research was that more complex and automated maneuvers have to be implemented and the testing method has to be improved (e.g. flying figures of eight to determine the turn rate). At the moment we have made great progress in terms of automation, improvement of the sensor system as well as maneuver enhancement, but this is the subject of current activities and can therefore not be further detailed.

**Referee:** *"the line sag is mentioned in the outlook, but shouldn't it be discussed in the error analysis (3.3.3), especially for static depowered flight?"*

I did not consider the line sag to minimize the post-processing. Since we use the same bar every test run, the line sag is similar for every kite, which means the repeatability can be clearly demonstrated with this method and the kites can be compared against each other (see p.6 "In order to facilitate an easy assessment of the measurement results as well as the reliability of the method, post-processing calculations to optimize the estimation of the properties were not carried out").

One of the central tasks of the subsequent work is the systematic investigation and improvement of the measuring method e.g. by performing post-processing calculations, which includes the line sag. We will incorporate this in the revised outlook.

**Referee:** *"are the errorbars in Figure 14 realistic as the $C_L$ coefficient depends on wind speed, for which an error of 20% is assumed (3.3.2)?"*

the error of wind speed is mentioned as $\delta v_{w,real} <= +20\%$ for a worst case scenario (wind vector $v_{tw}(z_{REF})$ pointing exactly towards the opposite direction of travel, a static wind velocity of $3m/s$ as well as an overestimated coefficient of friction $\alpha$), whereas at the presented day the wind conditions where much better.

[Figure]

**3. Technical corrections**

Thanks for the advice, we'll rework it.

[Figure]

**Automatic Measurement and Characterization of the Dynamic Properties of Tethered Membrane Wings**

Jan Hummel[1], Dietmar Göhlich[1], and Roland Schmehl[2]

[1]Methods for Product Development and Mechatronics, Technische Universität Berlin, 10623 Berlin, Germany
[2]Faculty of Aerospace Engineering, Delft University of Technology, 2629 HS Delft, Netherlands

**Correspondence:** Jan Hummel (jan.hummel@tu-berlin.de)

**Abstract.** ~~The performance of an airborne wind energy system crucially depends on the aerodynamic, structural dynamic and flight dynamic properties of the tethered wing. In the scope of this paper, flight dynamic properties are determined against reproducible steering inputs. Current design methods for highly flexible membrane wings have achieved a mature product level by combining iterative testing with empirical and intuitive variation of wing parameters. However, for significant further improvements, experimental data of high quality is indispensable.~~ We have developed a tow test setup for reproducible measurement of the dynamic properties of different types of tethered membrane wings. The test procedure is based on repeatable automated maneuvers with the entire kite system under realistic conditions. By measuring line forces and line angles, we determine the aerodynamic coefficients and the lift-to-drag ratio as functions of the length ratio between power and steering lines. This non-dimensional parameter characterizes the angle-of-attack of the wing and is varied automatically by the control unit on the towed test bench. During each towing run, several test cycles are executed such that mean values can be determined and errors can be minimized. We can conclude from this study that an objective measurement of specific dynamic properties of highly flexible membrane wings is feasible. The presented tow test method is suitable for quantitatively assessing and comparing different wing designs. The method represents an essential milestone for the development and characterization of tethered membrane wings as well as for the validation and improvement of simulation models. On the basis of this work, more complex maneuvers and a full degree of automation can be implemented in subsequent work. It can also be used for aerodynamic parameter identification.

**1 Introduction**

With the turn of the millennium, kitesurfing has evolved into a mainstream water sport, followed by the more recent variants of land and snow kiting (Tauber and Moroder, 2013). In terms of industrial applications, flexible membrane wings have already been used since the 1970s as aerodynamic decelerators for airdrop systems and are currently being explored for airborne wind energy (AWE) generation (Schmehl, 2018). Despite the advancements within the kitesurfing and AWE industries, tethered membrane wings are mostly still designed by iterative testing with empirical and intuitive variation of wing parameters. Although this has led to a relatively high degree of maturity on product level, the approach is time consuming and expensive because a large number of prototypes need to be manufactured and tested. For this reason, we conclude that the empirical

[Figure]

**Figure 1.** left: Leading Edge Inflatable (LEI) tube kites (single-skin kite); right: ram air wing

[revised manuscript text omitted]
 ~~wings was not feasible. However the knowledge of these properties is essential to evaluate existing simulation models as well as to characterize wings of AWE systems. As demonstrated in Sect. 5, the presented work fills this research gap. A unique test bench was developed to record these properties. It has been shown that a reproducible measurement of tethered flexible wings is possible. The repeatability was demonstrated by 8 recorded measurement files executing the~~ membrane wings exceeds the available budget. Furthermore, existing approaches do not allow for a recording or even automation of steering inputs, which is crucial for the reproducibility of the experiment. In this paper, we have presented a unique tow test setup for automatic measurement of the dynamic properties of different wing types at full scale and under realistic conditions. The objective was to

demonstrate the methodology and particularly the repeatability of the test procedure. Using the maneuver "Linear Power"  ", we determine the aerodynamic coefficients and lift-to-drag ratio of the wing as functions of the ratio of power and steering

5 line lengths – denoted as relative power setting – by measuring line forces and line angles. The ratio is varied automatically, while the pilot is manually adjusting the steering line lengths to keep the ~~wing at the zenith position within the wind window. Differences between the measured kites as well as reliability could be shown. In order to improve the measuring accuracy, it is necessary to develop the test bench further. The accuracy of the airflow measurement can be greatly increased by measuring at wing height. In addition, the determination of the airflow direction could be implemented. This should greatly improve~~

10 ~~the accuracy in the calculation of aerodynamic coefficients. The identification of the true angle of attack can be noted as an additional option for the improved determination of the aerodynamic efficiency as well as the aerodynamic components. As mentioned above, however, a high development demand is assumed to be required for this purpose, as it is difficult to identify the angle of attack for fully flexible wings . Additionally, further improvements could be done in measurement data evaluation. For example, the quality of the measurement results can be increased by calculating the line sag and the influence of weight.~~

15  kite at a fixed position relative to the towing vehicle. By automating the test cycles we can acquire mean values of high statistical quality with minimal errors. We have demonstrated the repeatability on the basis of eight recorded data sets using the maneuver "Linear Power" at a constant wind speed of 22 kn (11.3 m/s). We conclude from this study that it is feasible to objectively measure the flight dynamic properties of tethered

20 membrane wings and to quantitatively assess and compare different wing designs.

Based on this work, we propose several functional enhancements for future research. By performing more sophisticated flight maneuvers the full operational envelope of AWE systems can be covered. By completing the automation of the process we expect a significant increase of measurement accuracy which will improve future aerodynamic parameter identification as well as

25 ~~were determined by the maneuver "Linear Power", which features automatic line length control between power and steering lines, combined with a manual control of the length of the steering lines. The maneuver has been executed within 4.5 seconds. This timespan was chosen on the assumption that the kite is in static equilibrium at all times of the maneuver, which has to be investigated in further systematic tests. Moreover, in order to increase reproducibility and perform aerodynamic parameter identifications, it is necessary to implement and automate more sophisticated maneuvers. However, it is now feasible to perform~~

30 ~~aerodynamic parameter identification, because of recording the steering inputs and to be able to fully automate maneuvers in future. Furthermore, the developed test bench can be used to compare simulation models with measurement data. Starting from a uniform wing position, for example at the zenith position, it is possible to execute automated control inputs over a certain period of time. In order to keep the resulting error as low as possible, this maneuver has to be measured several times. The control inputs and airflow velocity can be used as input values for the simulation. The calculated positions and forces can be~~

[revised manuscript text omitted]